# A Non-Probiotic Fermented Soy Product Reduces Total and LDL Cholesterol: A Randomized Controlled Crossover Trial

**DOI:** 10.3390/nu13020535

**Published:** 2021-02-06

**Authors:** Sarah M. Jung, Ella H. Haddad, Amandeep Kaur, Rawiwan Sirirat, Alice Y. Kim, Keiji Oda, Sujatha Rajaram, Joan Sabaté

**Affiliations:** Center for Nutrition, Healthy Lifestyle & Disease Prevention, School of Public Health, Loma Linda University, Loma Linda, CA 92350, USA; smjung@students.llu.edu (S.M.J.); akaur1@llu.edu (A.K.); rsirirat@llu.edu (R.S.); alkim@students.llu.edu (A.Y.K.); koda@llu.edu (K.O.); srajaram@llu.edu (S.R.); jsabate@llu.edu (J.S.)

**Keywords:** fermented soy powder, total cholesterol, LDL cholesterol, Q-Can natural, isoflavones

## Abstract

Traditional Asian fermented soy food products are associated with reduced cardiovascular disease risk in prospective studies, but few randomized controlled trials have been conducted in at-risk populations. The aim of this study was to investigate the effect of a commercial non-probiotic fermented soy product on blood lipids in adults with cardiovascular risk biomarkers. In a randomized, crossover, intervention study, 27 men and women (aged 29–75 y) exhibiting at least two risk factors, consumed two packets (12.5 g each) daily of a fermented powdered soy product, or an isoenergic control powder made from germinated brown rice for 12 weeks each. The consumption of the fermented soy product resulted in a significantly greater mean change from baseline (compared to the germinated rice, all *p* < 0.05) in total cholesterol of −0.23 mmol/L (CI: −0.40, −0.06) compared with 0.14 mmol/L (CI: −0.03, 0.31), respectively; and low density lipoprotein (LDL) cholesterol −0.18 mmol/L (CI: −0.32, −0.04) compared with 0.04 mmol/L (CI: −0.01, 0.018) respectively. This was accompanied by an increase in high density lipoprotein (HDL) cholesterol in the germinated rice group, a decrease in apolipoprotein B (ApoB) in the fermented soy group, and a between-treatment effect in apolipoprotein A1 (ApoA1); however, the ratio of the LDL:HDL and of Apo B:ApoA1 did not differ between the groups. The ratio of total cholesterol:LDL decreased in men in the fermented soy group (*p* < 0.001). Twenty-four-hour urine collection at the end of each treatment period resulted in an increased excretion expressed as a ratio in μmol/d between treatments of 10.93 (CI: 5.07, 23.54) for daidzein; 1.24 (CI: 1.14, 4.43) for genistein; and, 8.48 (CI: 4.28, 16.80) for glycitein, all *p* < 0.05. The fermented soy powder consumed by participants in this study without implementing other changes in their typical diets, decreased the total and LDL cholesterol, and may serve as a dietary strategy to manage blood lipids. The trial was registered at ClinicalTrials.gov as NCT03429920.

## 1. Introduction

Despite significant advances in scientific medicine, cardiovascular disease (CVD) continues to be a major cause of ill health for men, women, and peoples of most racial and ethnic groups in the US and elsewhere [1]. Since its prevention and initial clinical management hinges on dietary modifications, identifying effective strategies to modulate dysregulated lipids is critical to reducing individual and population disease burden. An increased consumption of plant-based foods has been associated with a reduced risk, and although soy foods are consumed worldwide and promoted as healthy, their cardiovascular benefits remain disputed. Prospective studies of the association between soy intake and CVD mortality have shown protection in some studies [2,3], whereas others did not [4,5].

Soybeans contain vegetable protein, soluble fibers, oligosaccharides, minerals and the bioactive phytoestrogens, particularly the isoflavones genistein and daidzein. They are low in saturated fat and a source of unsaturated and omega-3 (n-3) fatty acids. As high-protein foods, the benefits of soy can occur as a consequence of soy displacing other protein foods in the diet such as saturated-fat containing meat and dairy [6], from the effect of the soy protein itself [7], or, from the effect of the other constituents of soy. The dietary intake of isoflavones in particular is associated with favorable CVD risk reductions in some population studies [8,9]. However, clinical trials with isoflavones have been inconsistent but more likely to show benefit when isoflavones were consumed as constituents of soy foods [10,11,12,13,14] rather than as supplements [15,16].

Many traditional forms of soy foods consumed by East Asian peoples are fermented products [17] and have been implicated in the health effects of soy among Asians [2]. Fermentation improves the functional qualities of the soybean by increasing its digestibility and isoflavone bioavailability [18]. However, few intervention trials have explored fermented soy foods and products for their lipid-lowering effects [18,19].

Prior to this, the study that investigated the effects of a commercial fermented soy product which does not contain active cultures on blood lipids in adults was a single arm before–after clinical trial in ten obese males and females [20]. In this crossover dietary intervention, we intended to examine the hypothesis that the daily consumption of 25 g of a related fermented soy powder would result in cardiometabolic benefits compared with an energy-matched comparison powder derived from germinated brown rice.

## 2. Materials and Methods

### 2.1. Study Design

The study was conducted as a randomized 2 × 2 controlled, crossover, dietary intervention study performed in two 12-week intervention periods with a 2-week washout period between periods (Figure 1). Participants were enrolled at 3 separate time points and randomly assigned to 2 study sequences, either fermented soy powder followed by germinated brown rice powder or vice versa in typical crossover fashion. A table of random numbers and a block stratification scheme that accounted for sex, age and body mass index (BMI) in kg/m^2^ were used to ensure randomization and balance between sequences.

An initial one-week run in period was allowed for instruction and adjustment to the research protocol. During each of the 12-week intervention periods, the participants attended clinics at baseline and at weeks 6, 11 and 12. Study materials were distributed and anthropometric and cardiometabolic endpoints were assessed by measurements and blood draws at clinics held at baseline and at weeks 11 and 12 of each intervention period. A clinic at week 6 was scheduled for participants to receive additional study supplement packets.

During both intervention periods, the participants received journals to record illness or deviations from their usual diet, medications, lifestyle, and experimental protocol. Dietary measurements were assessed from 4 unannounced telephone 24 h dietary recalls obtained from participants. Two 24 h recalls were obtained during each intervention period, one for a week day and one for a weekend day. Dietary intake data were collected and analyzed using the Nutrition Data System for Research software, version 2018, developed by the Nutrition Coordinating Center, University of Minnesota, (Minneapolis, MN, USA).

The study was conducted at the Nutrition Research Center of Loma Linda University (Loma Linda, CA, USA) between September 2018 and August 2019. The study protocol and relevant documents were approved by the Institutional Review Board of Loma Linda University before the initiation of the trial, and the participants provided written informed consent before enrolling in the study. The study was conducted according to the ethical study guidelines of the Declaration of Helsinki of 1975 as revised in 1983. The trial was registered at ClinicalTrials.gov as NCT03429920.

### 2.2. Participants

Eligible participants were adult men and women aged 30–75 y inclusive, with at least 2 risk factors for CVD. Participants were recruited from employees of Loma Linda University (Loma Linda, CA, USA) and from the general population surrounding the institution, using volunteer databases from previous studies, posters and social media announcements. A total of 29 participants were randomized but 2 participants dropped out early for lack of interest: one before any baseline measurements were obtained and the other with partial baseline measurements (Figure 2).

The considered CVD risk factors were: smoking; systolic blood pressure ≥ 140/90 mmHg; low density lipoprotein (LDL) cholesterol ≥ 110 mg/dL; high density lipoprotein (HDL) cholesterol ≤ 40 mg/dL; triglycerides ≥ 150 mg/dL; fasting blood glucose ≥ 110 mg/dL; BMI ≥ 25; and, family history of heart disease. Exclusion criteria were: uncontrolled renal, hepatic, or endocrine disease assessed from participant’s medical history and clinical laboratory data; use of lipid-lowering drugs (statins), or supplements containing plant sterols/stanols; hypersensitivity or allergy to soy or brown rice; alcohol or drug addiction or abuse; insulin-dependent diabetes; lack of ability or interest to follow the study protocol. Also excluded were pregnant or lactating women or those with childbearing potential and unwillingness to commit to the use of a medically approved form of contraception throughout the study period. Participants were instructed to maintain their usual physical activity levels and consistent lifestyle habits during the study period.

To establish eligibility, the participants were first screened based on a self-completed online health and lifestyle questionnaire and the results of a preliminary medical history and standard chemistry and hematology profile. The eligible participants were invited to attend an informational group session at the Nutrition Research Center when the study protocol was thoroughly explained. The participants were interviewed by researchers and only those deemed eligible and who demonstrated their willingness to participate were randomized into the study.

### 2.3. Intervention Products and Compliance

In this study, the participants were required to consume 2 packets of the intervention products daily for 24 weeks. The 2 types of intervention foods were isocaloric and matched for color, flavor and textures and provided in powder form. Table 1 compares their nutrient composition. Each fermented soy packet was 12.5 g of Q-Can Natural powder and the comparison packet was an isoenergic amounts of dried germinated brown rice powder, approximately 15 g. Both products were similarly flavored with chocolate flavor and monk fruit sweetener and packaged by the research sponsor Beso Biological Research, Inc., (Diamond Bar, CA, USA). Participants were instructed to consume the powder by mixing it with 8 fluid ounces of a beverage of their choosing such as water, non-fat milk, plant-based milk and to remain consistent with the same beverage throughout the study both for the fermented soy and germinated rice periods.

The soy powder investigated in this study was a patented food produced from organic soybeans which have been fermented with a guarded confidential mix of microorganisms and is available commercially as Q-Can Natural. The product is not a probiotic as it does not contain live cultures.

Compliance with intervention product intake was calculated from returned wrappers and unused packets. It was also deduced from the participants’ unsolicited mention of the intervention product consumption during the unannounced dietary 24 h recalls. In addition, isoflavone assays of 24 h urine collections provided an indication of group compliance.

### 2.4. Anthropometric and Laboratory Measurements

Height, weight, and waist circumference were collected by trained research clinicians on 6 occasions, at baseline and at weeks 11 and 12 of each of the 2 intervention periods. Body weight was measured to the nearest 100 g using a Tanita Body Composition Analyzer (TBF300A) (Arlington Heights, IL, USA). Height was measured using a wall mounted Detecto stadiometer (Webb City, MI, USA) to the nearest 0.1 cm, with the participant in an erect position, without shoes, and with head, shoulder blades, hip, and heels touching the measurement surface. Waist circumference was measured horizontally 1 cm above the navel to the nearest 0.1 cm. Measurements were performed under normal settings with participants wearing light loose clothing. Blood pressure was recorded using an ambulatory blood pressure monitor. Participants were relaxed, seated in a quiet, light room for 15 min before the commencement of the measurement, which lasted for 15 min and included 3 measurements.

All blood samples were collected after an overnight fast (≥10 h). Baseline blood samples were collected at day 0 of each intervention period, and post-intervention samples were collected on 2 separate days at weeks 11 and 12 of the period. After separation by centrifugation, serum and plasma were aliquoted and stored immediately at −80 °C until the project was completed. Samples for blood lipids, apolipoproteins, glucose and insulin were shipped frozen to and analyzed at the Nutrition Evaluation Laboratory, John Mayer USDA Human Research Center at Tufts University (Medford, NJ, USA). Tests per participant were assayed in the same analytical run to reduce systematic error and inter-assay variability. Concentrations of blood lipids and glucose were measured using an automatic clinical chemistry analyzer Olympus AU480 with appropriate enzyme-based reagents and controls (Beckman Coulter, Inc., Brea, CA, USA). The intra- and inter-assay % coefficient of variation (CV) for total cholesterol, triglycerides, HDL cholesterol, and glucose were 2.0 and 2.8%, 2.0 and 3.4%, 3.0 and 5.0%, and 2.0 and 3.4%, respectively. The Friedewald formula (LDL cholesterol = total cholesterol − HDL cholesterol − triglycerides/2.9) was used to estimate LDL cholesterol when triglyceride levels were less than 400 mg/dL, otherwise LDL cholesterol was determined directly on the AU480 with intra- and inter-assay CVs of 2.4 and 3.6%, respectively. ApoA1 and ApoB were measured on the Olympus AU480 using an immune-turbidimetric procedure with an average intra- and inter-assay %CV of less than 2.5 and 3.5%, respectively. Serum insulin was measured by solid-phase, two-site chemiluminescent immunometric assays using the IMMULITE 2000 (Siemens Healthcare Diagnostics, Los Angeles, CA, USA) with an intra- and inter-assay CVs of 3.5 and 5.0%, respectively. Homeostatic Model Assessment for Insulin Resistance (HOMA_IR) was calculated as HOMA-IR = insulin (microU/L) × glucose (nmol/L)/22.5.

Twenty four-hour urine was collected once following each intervention at the end of the period. For the collection of the urine, the volunteers were provided with a 3-L collection container, ice packs and an isothermic bag and were instructed to start collecting the sample the morning before the study visit and for 24 h. Once the urine was received in the lab, urine volume was recorded. The pH aliquots were adjusted to 4 with 1 M hydrochloric acid and stored at −80 °C until further analysis. Urine samples were shipped frozen to Creative Proteomics (Shirley, NY, USA) where they were quantified for isoflavones (genistein, daidzein, glycitein) using an AB SCIEXAPI tandem mass spectrometer (MS) connected with a Waters Acquity (Milford, MA, USA) ultra-performance liquid chromatographer (UPLC) An internal standard mix was added to the urine, followed by a β-glucuronidase solution. Mixtures were incubated overnight at 37 °C and extracted by hexane and either. The extracted material was dried under nitrogen and reconstituted with 80% methanol. After centrifugation, the supernatant was taken up for UPLC–MS/MS analysis.

### 2.5. Statistical Analysis

The primary outcome of this study was to measure relative changes in LDL cholesterol and it was calculated that 21 participants were needed to detect a 10% change (α = 0.05 and 80% power) in LDL cholesterol assuming a 20 mg/dL within-subject SD and a correlation of 0.4 between the treatment and control phases of the study [21]. To allow for dropouts, 29 participants were recruited and the data for 27 individuals were used for statistical analysis.

Unless stated otherwise, the tests of significance were 2-sided and were performed at a 5% level of significance. The baseline comparability of the intervention-sequence groups was assessed by *t*-test, Fisher’s exact test and Mann–Whitney test as appropriate. The primary outcome variable was the mean change from baseline (day 0) to the end of each intervention period in LDL cholesterol concentration, with both week 11 and week 12 values used as post-intervention measurements. Other outcome variables included the changes from baseline were: BMI; total cholesterol; HDL cholesterol; triglycerides; total cholesterol:HDL cholesterol ratio; LDL:HDL ratio; ApoA1; ApoB; ApoB:ApoA1 ratio; glucose; insulin; and HOMA-IR. HOMA-IR was calculated as follows: Insulin (μU/mL) × Glucose (mg/dL)/405. For each of the outcomes, a mixed model was fitted to compare the changes from baseline to the end-of-study between the study period. The mixed model included: treatment (fermented soy, germinated brown rice); time (baseline, end-of-study); an interaction term for treatment and time; sequence (1, 2); visit within each phase (baseline, week 11, week 12); and enrollment periods (1, 2, 3) as fixed-effects terms; and participants as random-effects term. To test if there was a significant change within treatment, a *p*-value of the simple effect of time was reported for each treatment. To test if the changes were significantly different between the two treatments, a *p*-value of the treatment-by-time interaction was reported for each outcome. Marginal means with 95% confidence intervals were estimated for each treatment and time combination. Assumptions of the mixed models were assessed using plots of conditional residuals. For variables with a skewed distribution, a log-transformation was applied then back-transposed to the original scale. For log-transformed outcomes such as glucose, insulin and HOMA-IR, the differences of log means were expressed as a mean ratio. Mixed models were also applied to urine isoflavones as described above but without terms for time and its interaction with sequence.

To examine whether the effect of the intervention differed between males and females, additional mixed models were fitted for each outcome, this time including gender in a 3-way interaction of treatment × time × sex (male, female). To examine whether the effect of the interventions differed among females due to menopausal status, additional mixed models were fitted for each outcome, this time including menopausal status in a 3-way interaction of treatment × time × menopause (yes, no).

All data were analyzed using SAS version 9.4 (SAS Institute, Cary, NC, USA) and R version 3.6.3.

## 3. Results

### 3.1. Participants, Dietary Intake, and Compliance

Of the 61 volunteers screened, 29 met the inclusion criteria and were randomly assigned to one of the following sequences: a fermented soy and control sequence (*n* = 12) or a control and fermented soy sequence (*n* = 17). Of the 27 participants who completed the study, 21 were women and six were men with an overall mean age of 51.6 y (SD 13.5 y) and a range of 29–75 y. All participants met inclusion criteria regarding cardiovascular risk factors, and at baseline, had mean values of: BMI = 32.5 kg/m^2^; total cholesterol = 5.22 mmol/L; and LDL cholesterol = 3.13 mmol/L. There were no statistically significant differences between the two sequences in any of the baseline measurements (Table 2). Neither intervention had an effect on BMI (Table 3), or on waist circumference, body fat percentage or blood pressure throughout the study period (data not shown). Based on data from the 24 h dietary recalls, no change in daily energy, or total dietary fiber, soluble fiber, fat, total protein, animal protein, plant protein, vitamin or mineral intake was reported by participants except for carbohydrate, cholesterol and isoflavones (Appendix A).

The returned unconsumed wrappers showed 75% compliance, whereas the 24 h recalls showed 90% compliance in reporting the consumption of the research foods. A measure of group compliance was shown by the higher concentrations of urine isoflavone (daidzein, genistein and glycitein) biomarkers of soy intake in the fermented soy group compared to the germinated brown rice group with *p* values < 0.001 for all (Figure 3 and Appendix A).

### 3.2. Laboratory Measures

Marginal means estimated from the mixed models for cardiovascular and metabolic markers at baseline and at the end of each diet are shown in Table 3. Compared with geminated rice, the fermented soy intervention significantly decreased the concentrations of LDL cholesterol and total cholesterol with statistically significant interaction effects of *p* < 0.05 and *p* < 0.01, respectively. The interventions resulted in a significant between-treatment effect in HDL cholesterol and an increase in HDL in the germinated rice group, but changes in the total cholesterol:HDL ratio and LDL:HDL ratio did not differ between interventions. There was also a significant between-treatment effect in ApoA1 (*p* < 0.05), and Apo B which showed a within-treatment decrease in the fermented soy group. However, changes in the ApoB:ApoA1 ratio did not differ between treatments. There were no significant changes to serum triglyceride, fasting glucose or insulin concentrations, or to the HOMA-IR index.

Examining the differences in the effects of the intervention powders between males and females did not show significant three-way interactions of treatment x time x sex for BMI (*p* = 0.4246), total cholesterol (*p* = 0.1140), LDL (*p* = 0.6811), HDL (*p* = 0.7784), triglycerides (*p* = 0.1709), glucose (*p* = 0.9598), insulin (*p* = 0.2521), HOMA-IR (*p* = 0.2015), ApoA1 (*p* = 0.9410), ApoB (0.8386), LDL:HDL ratio (0.4250) or the ApoB:ApoA1 ratio (*p* = 0.4781). However, there was a significant three-way interaction in the total cholesterol:HDL ratio (*p* = 0.0014) indicating the effect modification of the variable by sex. Marginal means for the total cholesterol:HDL ratio for males and females are shown in Table 4.

Among the 21 women in the study, 11 indicated they were post menopause. To examine whether the effects of the study interventions differed between pre- and post-menopause women, additional mixed models were fitted for each outcome, this time with treatment × time × menopause (yes, no) and only in women. The three-way interaction was not significant in any of the outcomes indicating that the menopausal status had no effect on the outcomes in female participants.

## 4. Discussion

In this randomized controlled dietary intervention study, we reported evidence that the consumption of 25 g per day of a fermented soy powder produces clinically relevant reductions in the total cholesterol and LDL cholesterol concentrations of approximately 7% each compared to the comparison with germinated brown rice powder. Although a concomitant increase in HDL cholesterol in the germinated rice group and a decrease in ApoB in the fermented soy group were observed, changes in the total cholesterol:HDL ratio, LDL:HDL ratio or the ApoB:ApoA1 ratio did not differ between treatments. However, among males, there was a decrease in the total cholesterol:LDL ratio in the fermented soy group (*p* < 0.001). These results are noteworthy from a preventive perspective considering that our participants consumed their habitual diets, and although elevated lipids were one of the inclusion criteria for the study, participants had relatively normal baseline total cholesterol (mean ~5.2 mmol/L) and LDL-cholesterol (mean ~3.0 mmol/L) concentrations. It has previously been demonstrated that greater cholesterol reductions may be expected with dietary modifications in individuals with hypercholesterolemia [21,22].

Formulated from soybeans, the product investigated in the present study was fermented—a process which can affect its physiological function and may provide metabolic outcomes which differ from those of soy. Soy protein and soy’s main bioactive isoflavone constituents have long been considered candidates to improve various cardiovascular-related endpoints. The role of soy foods in cardiovascular risk derives from epidemiological observations and clinical trials. Prospective studies on the association between soy food consumption and CVD mortality have shown an inverse association [23,24,25]. However, the evidence is equivocal as in some studies the association was significant only in women [2,3], or with soy foods traditionally consumed in Eastern societies [26]. A prospective study on the association between soy-derived isoflavone intake and CVD reported reduced risk of myocardial and cerebral infarctions primarily among postmenopausal women [3]. However, the commonly consumed soy foods in Western societies contain relatively low levels [17], as a recent secondary analysis from three large prospective US cohorts reported that higher intakes of soy isoflavones and tofu were associated with a moderately lower risk of developing coronary heart disease both in men and women [27].

With respect to clinical intervention trials, studies early on found that substituting soy protein for animal protein lowers blood cholesterol concentration and the risk of CVD [28,29,30]. The results of a landmark meta-analysis on the lipid-lowering effects of soy in hypercholesterolemic individuals [21] led to an unqualified health claim in 1999 relating soy protein consumption to a reduced risk of heart disease [31]. However, subsequent controlled clinical trials on soy and soy-derived isoflavones determined that the impact of soy consumption on LDL cholesterol was inconsistent and smaller than initially reported [16]. On the basis of data from clinical trials, the US Food and Drug Administration stated that it is reconsidering its soy health claim. Nonetheless, a number of recent meta-analyses of studies on soy food intake and serum lipids demonstrated reductions in LDL cholesterol in the range of 3–4% [22,32,33].

In the current study, the fermented soy product tested was relatively rich in bioavailable isoflavone, as evidenced by the significantly higher urinary excretion of daidzein, genistein and glycitein on the fermented soy intervention. Clinical trials with isoflavone preparations have been equivocal and have underscored differences in efficacy between natural soy foods rich in isoflavones, compared to isoflavone supplements in reducing lipid biomarkers. It has been suggested that the lipid-lowering benefits of soy are better achieved when isoflavones are eaten within a soy product. Although some human trials demonstrated a hypolipidemic effect when isoflavones are consumed independently of soy foods or soy protein [34,35], others have not [22,36]. Furthermore, findings from intervention trials with isoflavone-rich soy foods have also been inconsistent. A trial in which 12.5 and 25 g protein from defatted soy flour rich in isoflavones was incorporated into muffins and consumed by hypercholesterolemic adults did not reduce LDL cholesterol or other risk factors [37].

Traditional Asian diets use fermentation to produce several types of soy foods such as tempeh, natto, chungkookjang and miso, and various microorganisms are involved in their production [38]. Fermentation modifies soybean properties in various ways such as in its digestibility, nutrient content, and isoflavone chemical configuration [18]. In a prospective study from Japan, a higher intake of fermented soy products was associated with a lower risk of CVD mortality in both men and women aged 45–75 years [39]. In another cohort, natto intake was significantly inversely associated with the risk of mortality from cardiovascular disease [26]. Natto, a soy product fermented with *Bacillus subtilis* contains a known antithrombotic fibrinolytic enzyme which may partly explain the beneficial effect of the fermented product [40].The Korean traditional chungkookjang improved body composition, lipid profiles and atherogenic indices when tested in overweight and obese participants [41].

The fermentation of soy foods enhances the bioavailability of its phytoesterogen components. Isoflavones in unfermented soy foods occur naturally as glycoside conjugates which are hydrolyzed to absorbable aglycones by intestinal microflora [42]. Fermentation hydrolyzes the glycosidic isoflavones in the food product itself and it has been demonstrated that the isoflavone aglycones in fermented foods are absorbed faster and in greater amounts than isoflavones from unfermented products [43]. Both traditional fermented soy foods [19,44] and commercially prepared fermented products [20,45] have reliably shown lipid-lowering effects in clinical trials.

Several biological explanations have been proposed to support the putative effect of soybeans and their constituents on cardiovascular risk. Soy foods are low in saturated fat, a source of omega-3 (n-3) fatty acids, fiber and bioactive substances, and their consumption may displace animal-source protein foods high in saturated fat and low in fiber [6]. Alternatively, the benefits of soy can occur as a consequence of the soy protein itself. One cholesterol-lowering mechanism of soy has been related to the 7S globulin fraction known to inhibit hepatic ApoB synthesis [46]. The protein peptide β-conglycinin found in soy reduced blood lipids in hyperlipidemic women [47]. It is possible that the production and bioavailability of various bioactive peptides is enhanced in fermented soy products. A number of bioactive peptides have been shown to inhibit hepatic lipid and cholesterol synthesis [48]. Two peptides isolated from soy β-conglycinin have properties similar to those of statins in that they competitively inhibit the hydroxymethylglutaryl–CoA reductase enzyme and inhibit cholesterol synthesis [49]. It has also been suggested that the biological processes by which soy isoflavones lower blood lipids and reduce the risk of cardiovascular disease may be estrogen related [50]. Considering that isoflavones are structurally similar to 17-β-estradiol, they have demonstrated estrogen-like activities, and have shown a high binding affinity to the beta-estrogen receptor (ERβ). ERβ is expressed in coronary vessels and their activation leads to many cardiovascular protective effects, primarily endothelial nitric oxide synthesis and vascular reactivity and vasodilation [51]. Isoflavones have important estrogen-like effects on the lipid profile and can modulate cholesterol metabolism promoting hepatic cholesterol catabolism [52].

In the current study, the isoflavone-rich fermented soy supplement did not influence fasting glucose or insulin concentrations or improve insulin sensitivity. Previous trials on the effect of soy consumption on glycemic control have been inconsistent. Although a meta-analysis of RCT studies indicated that soy isoflavone supplementation may improve glucose metabolism in non-Asian postmenopausal women [53], other randomized clinical trials failed to demonstrate that soy protein with or without isoflavones, or the isoflavones daizein and genistein, had favorable effects on glycemic control or insulin sensitivity [54,55].

The increase in HDL cholesterol concentrations in the germinated brown rice group was unexpected and notable. Recently, similar germinated rice products have also shown increased HDL effects in rats [56] and in individuals with metabolic syndrome [57]. The fact that fecal cholesterol excretion increased significantly with germinated brown rice suggests a possible mechanism related to the fiber or polyphenol content of brown rice [56].

By employing a crossover design in this study in which participants served as their own control, we were able to decrease potential sources of error by minimizing the influence of between-subject variability when analyzing treatment effects. In addition, to minimize within-subject variability, blood draws and measurements were collected two times at the end of the intervention period approximately one week apart. Although generally healthy with a relatively wide age range, our participants were at risk for cardiovascular disease since they were recruited to exhibit a minimum of 2 risk factors to enroll. The 24 h recalls and returned wrappers indicated that subjects consumed the study powders and were generally adherent to the instructions on their use.

One weakness of this study was the much larger proportion of females to male participants. Although isocaloric and similar in appearance, the powders differed in macronutrient composition which may have confounded the outcomes. In addition, the sharp odor of the fermented soy supplement may have impacted compliance in some participants. Moreover, the germinated brown rice powder may not have been an entirely neutral product and may have had physiological properties that influenced the results as explained above. Another major weakness is that the participants’ diets were not restricted and differences in habitual intakes might have occurred between treatments such as in the intake of phytonutrients and micronutrients. As expected, the 24 h recalls show an increased intake of carbohydrate on the germinated rice intervention, however, the increase in dietary cholesterol cannot be explained by differences in the composition of the intervention products and suggests critical dietary alterations which may have influenced the outcomes. A possible explanation for the variation in cholesterol intake is the seasonality of food group consumption [58]. Although a crossover design was employed, participants were recruited in phases and the study spanned multiple seasons.

The consumption of fermented soy foods may hold promise in lipid management. Current recommendations for cardiovascular risk reduction promoted the greater intake of plant-based foods. The use of functional foods and natural products can provide health benefits beyond nutrition and their use is popular among consumers [59]. However, the effectiveness of many such products is dubious and merely anecdotal. Although clinical trials are few, the evidence supporting fermented soy products for cardiovascular risk reduction has been somewhat consistent and warrant additional studies using both probiotic and non-probiotic fermented soy foods.

In conclusion, this study demonstrates that the daily consumption of two packets of a non-probiotic fermented soy product as part of one’s habitual diet by men and women who at risk for heart disease may result in reductions in total cholesterol and LDL cholesterol in both men and women, and in the total cholesterol: LDL ratio in men. The hypolipidemic effect of fermented soy may be mediated through a synergistic interaction between soy protein, its phytoestrogens and other soy components and may be enhanced by process of microbial fermentation. Future studies are needed to confirm these findings and to further investigate the biological processes that contribute to these effects.

## Figures and Tables

**Figure 1 nutrients-13-00535-f001:**
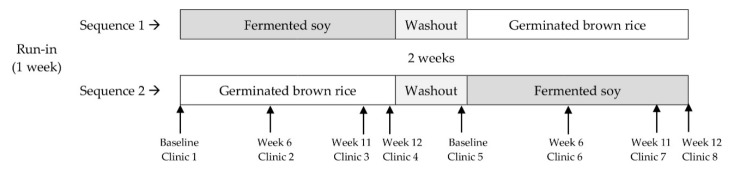
Study design of a randomized, controlled, crossover study in which participants with at least two risk factors for cardiovascular disease (CVD) consumed daily 2 packets of a powdered fermented soy product Q-Can Natural or 2 packets of a powdered germinated rice food, each dissolved in a liquid for a period of 12 weeks each, separated by a 2-week washout period. The packets were taken in a beverage of their choice—either water, low-fat milk or a plant-based milk—but participants had to be consistent in their choice of beverage. Anthropometric and blood pressure measurements and fasted blood were collected on study visits at baseline and weeks 11 and 12. A 24 h urine sample was collected at week 12.

**Figure 2 nutrients-13-00535-f002:**
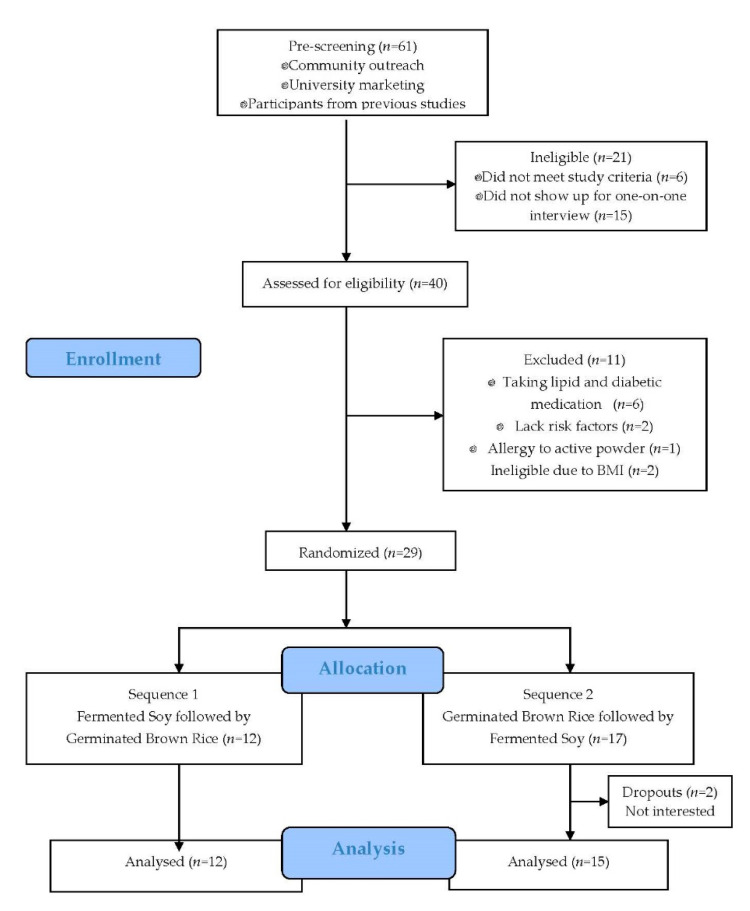
Consolidated standards of reporting trials diagram indicating sample sizes at each stage during the study.

**Figure 3 nutrients-13-00535-f003:**
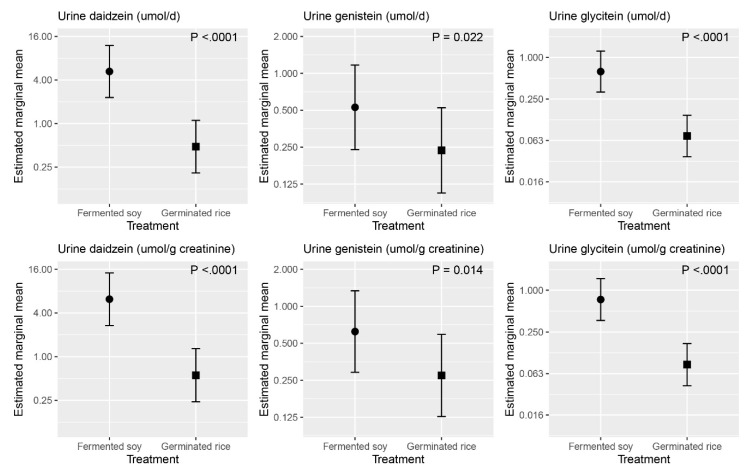
Estimated marginal means for the 24 h urine excretion of isoflavones (daidzein, genistein, glycitein) following the fermented soy and germinated rice intervention. Note: The *y* axis on a log_10_ scale.

**Table 1 nutrients-13-00535-t001:** Nutrient composition of study foods.

	Fermented Soy ^1^ 2 Packets (25 g)	Germinated Brown Rice ^2^ 2 Packets (30 g)
Energy, kcal	110	110
Carbohydrate, g	8.4	23.0
Protein, g	9.4	2.2
Total fat, g	4.54	0.84
Saturated fat, g	1.04	0.16
Monounsaturated fat, g	1.30	0.30
Polyunsaturated fat, g	2.20	0.30
Linolenic acid 18:3, g	3.4	0.03
Dietary fiber, g	5.6	1.4
Total isoflavones, mg	36.3	0

^1^ Nutrient composition provided by Beso Biological Research, Inc. (Diamond Bar, CA, USA). ^2^ Nutrient composition from US Department of Agriculture, Agricultural Research Service, Food Data Central, 2019.

**Table 2 nutrients-13-00535-t002:** Participant characteristics at baseline for the 27 adults with at least 2 risk factors for CVD in the evaluable sample according to randomized group assignment ^1^.

	Sequence 1 Fermented Soy ⟶ Germinated Brown Rice	Sequence 2 Germinated Brown Rice ⟶ Fermented Soy	*p*-Value
	(*n* = 12)	(*n* = 15)	
Age, y	50.3 (12.3)	52.5 (14.8)	0.68^2^
Sex, *n* (%)			
Female	8 (66.7)	13 (86.7)	
Male	4 (33.3)	2 (13.3)	0.36 ^3^
Menopause, *n* (%)			
No	3 (37.5)	7 (53.8)	
Yes	5 (62.5)	6 (46.2)	0.659 ^1^
Waist circumference, cm	107.6 (22.3)	107.4 (13.0)	0.97 ^2^
Systolic blood pressure, mmHg	123 (16)	118 (20)	0.46 ^2^
Diastolic blood pressure, mmHg	80.5 (15.7)	73.1 (9.9)	0.15 ^2^
Total cholesterol, mmol/L	5.01 (0.80)	5.17 (1.33)	0.73 ^2^
LDL cholesterol, mmol/L	3.03 (0.68)	3.07 (1.17	0.93 ^2^
HDL cholesterol, mmol/L	1.36 (0.31)	1.41 (0.30)	0.71 ^2^
Triglycerides, mmol/L	1.35 (0.56)	1.51 (0.36)	0.53 ^4^
Glucose, mmol/L	5.87 (1.22)	5.51 (0.36)	0.83 ^4^
Insulin, mU/L	19.5 (13.8)	13.4 (6.8)	0.31 ^4^
HOMA-IR	5.34 (3.97)	3.26 (1.66)	0.31 ^4^
Apolipoprotein A1, g/L	1.36 (0.19)	1.43 (0.26)	0.45 ^2^
Apolipoprotein B, g/L	0.95 (0.17)	0.96 (0.29)	0.90 ^2^

^1^ Values are the mean + SD or *n* (%). ^2^ Two-sample *t*-test. ^3^ Fisher’s exact test. ^4^ Mann–Whitney test.

**Table 3 nutrients-13-00535-t003:** The BMI, blood lipids and metabolic markers of participants on fermented soy and control intervention at baseline, end of study and change from baseline.^1^ (*n* = 27).

	Fermented Soy				Germinated Brown Rice				
	Baseline (95% CI)	End of Study (95% CI)	Change within Treatments (95% CI)	Within *p*-Value ^2^	Baseline (95% CI)	End of Study (95% CI)	Change within Treatment (95% CI)	Within *p*-Value ^2^	Between *p*-Value ^3^
BMI, kg/m^2^	33.3 (30.1, 36.5)	33.0 (29.8, 36.3)	−0.27 (−0.57, 0.04)	0.0874	33.1 (29.9, 36.3)	33.2 (30.0, 36.4)	0.10 (−0.20, 0.41)	0.5073	0.0944
Total cholesterol, mmol/L	5.27 (4.73, 5.82)	5.04 (4.50, 5.58)	−0.23 (−0.40, −0.06)	0.0073	5.06 (4.51, 5.60)	5.20 (4.66, 5.74)	0.14 (−0.03, 0.31)	0.1004	0.0024
LDL cholesterol, mmol/L	3.15 (2.66, 3.64)	2.97 (2.48, 3.46)	−0.18 (−0.32, −0.04)	0.0132	2.96 (2.47, 3.46)	3.00 (2.51, 3.49)	0.04 (−0.10, 0.18)	0.5836	0.0317
HDL cholesterol, mmol/L	1.46 (1.31, 1.61)	1.43 (1.28, 1.58)	−0.03 (−0.09, 0.03)	0.2577	1.39 (1.24, 1.54)	1.48 (1.33, 1.64)	0.09 (0.03, 0.15)	0.0026	0.0036
Total cholesterol:HDL	3.76 (3.24, 4.28)	3.66 (3.15, 4.17)	−0.10 (−0.22, 0.02)	0.0976	3.76 (3.24, 4.27)	3.70 (3.19, 4.22)	−0.05 (−0.17, 0.06)	0.3568	0.5942
LDL:HDL	2.27 (1.83, 2.71)	2.17 (1.73, 2.61)	−0.10 (−0.20, 0.01)	0.0686	2.22 (1.78, 2.67)	2.16 (1.72, 2.60)	−0.07 (−0.17, 0.04)	0.2066	0.6790
Triglycerides, mmol/L	1.47 (1.18, 1.76)	1.52 (1.23, 1.81)	0.05 (−0.11, 0.20)	0.5339	1.54 (1.25, 1.83)	1.60 (1.31, 1.90)	0.06 (−0.09, 0.22)	0.4237	0.9017
Apo A1, g/L	1.46 (1.36, 1.56)	1.42 (1.32, 1.52)	−0.04 (−0.09, 0.01)	0.1267	1.44 (1.34, 1.54)	1.48 (1.38, 1.58)	0.04 (−0.01, 0.09)	0.1548	0.0390
Apo B, g/L	0.97 (0.86, 1.09)	0.94 (0.82, 1.05)	−0.04 (−0.07, −0.00)	0.0303	0.09 (0.08, 1.05)	0.09 (0.08, 1.05)	0.00 (−0.03, 0.04)	0.8900	0.0997
Apo B:ApoA1	0.69 (0.58, 0.80)	0.67 (0.57, 0.78)	−0.02 (−0.05, 0.01	0.2064	0.67 (0.57, 0.78)	0.66 (0.56, 0.77)	−0.01 (−0.04, 0.02)	0.4119	0.7504
			Change as Mean Ratio within Treatments				Change as Mean Ratio within treatments		
Glucose, mmol/L	5.79 (5.37, 6.24)	5.97 (5.53, 6.44)	0.06 (0.05, 0.06)	0.1099	5.83 (5.41, 6.29)	6.01 (5.57, 6.49)	0.06 (0.05, 0.06)	0.1088	0.9963
Insulin, mU/L	12.7 (9.11, 17.8)	12.4 (8.9, 17.2)	0.97 (0.81, 1.17)	0.7522	12.5 (8.88, 17.5)	12.7 (9.07, 17.7)	1.02 (0.84, 1.23)	0.8450	0.7177
HOMA-IR	3.27 (2.25, 4.74)	3.21 (2.23, 4.62)	0.98 (0.82, 1.17)	0.8333	3.23 (2.21, 4.70)	3.42 (2.37, 4.93)	1.06 (0.89, 1.26)	0.5164	0.5435

^1^ Values are the marginal means estimated for each treatment and time combination and a 95% confidence interval. ^2^
*p*-values of simple effect of time (baseline to end of study) for each treatment. ^3^ Treatment × time interaction *p*-values. Apo A1 = apolipoprotein A1; Apo B = apolipoprotein B; HOMA-IR = homeostasis model assessment-estimated insulin resistance.

**Table 4 nutrients-13-00535-t004:** Effect modification by sex on the total cholesterol:HDL cholesterol ratio ^1^.

	Fermented Soy				Germinated Brown Rice				
	Baseline (95% CI)	End of Study (95% CI)	Change within Treatments (95% CI)	Within *p*-Value ^2^	Baseline (95% CI)	End of Study (95% CI)	Change within Treatment (95% CI)	Within *p*-Value ^2^	Between *p*-Value ^3^
Female (*n* = 21)	3.42 (2.89, 3.96)	3.41 (2.88, 3.94))	−0.01 (−0.15, 0.12)	0.8320	3.43 (2.89, 3.96)	3.36 (2.83, 3.89)	−0.07 (−0.20, 0.07)	0.3228	0.6783
Male (*n* = 6)	4.81 (3.90, 5.72)	4.41 (3.51, 5.31)	−0.40 (−0.65, −0.15)	0.0019	4.81 (3.89, 5.72)	4.77 (3.86, 5.67)	−0.04 (−0.30, 0.23)	0.7719	0.0004

^1^ Values are the marginal means estimated for treatment and time for each sex and a 95% confidence interval. ^2^
*p*-values of simple effect of time (baseline—end of study) for each treatment. ^3^
*p*-values for treatment × time interaction for each sex.

## Data Availability

The data presented in this study are available on request from the corresponding author. The data are not publicly available due to ethical restrictions.

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
