# Peer review of "A Non-Probiotic Fermented Soy Product Reduces Total and LDL Cholesterol: A Randomized Controlled Crossover Trial"

_nutrients, 2021, doi:10.3390/nu13020535_

Round 1

Reviewer 1 Report

Thank you for the opportunity to review. This is an overall interesting and rigorously conducted study, but the report could benefit from edits:

A major flaw is the discrepancy between the reported primary outcome on clinicaltrials.gov and in the manuscript. Clinicaltrials.gov states that the primary outcome is a composite measure of serum lipids, while the manuscript states that the primary outcome is LDL-c. The primary report of this clinical trial should report on the registered outcome. Thus, the methods section and power calculations may be misleading here. In line with this recommendation, the title of the manuscript should be edited so that it is not cherry-picking statistically significant findings from the results.

Other comments

The manuscript switches between the use of "subjects" and "participants." I recommend using "participants" throughout.

Was urinary equol measured, or was equol producer status determined?

Minor comments:

Line 117-18: this statement belongs in the results section and not in the methods.

Lines 122-123: Spell out abbreviations at first use.

Line 124: how were uncontrolled diseases defined? e.g. by clinical values, self-report, etc?

Reviewer 2 Report

The authors of this RCT evaluate the effects of soy commercial fermented  supplementation versus an  isoenergic control powder made from germinated brown rice for 12 weeks each  (placebo) on plasma lipid levels and inflammatory markers in subjects with at least 2 cardiovascular risk factors. They find that the supplement with fermented derivative of soy significantly reduces the levels of LDL cholesterol associated with an increase in the active phytoestrogens of its components. However, HDL cholesterol levels are significantly increased in the control group.

Major comments:  Some methodological aspects need to be improved

-In the RCT registry they hope to randomize 31 subjects, but they only randomize 29 and finish the study 27. Does it reduce the statistical power of the study?

- But fundamentally they do not refer to how the sample size was estimated. What was the expected change in lipid levels? What was the variable selected for the estimation of the sample size? How powerful is the study to detect these changes? Was the type 1 and type 2 statistical error considered? This section must be made explicit

Minor comments:

- Adherence to the intervention is confirmed with the 24-h urine levels of isoflavones. What change did they hope to find? Does the differences found between the two groups justify that they consumed the 2 packages of 25 grams ?. Please explain

- The CV risk factors used are not those commonly used. They did not distinguish by gender in relation to the HDL level (40/50 male / female)? Fasting plasma glucose levels were 110 mg / dl instead of the more commonly used 100 mg / dl. Please explain.

- Considering the decrease in the LDL cholesterol level in the experimental group but the increase in the HDL levels in the control group, they should relate the HDL / LDL ratio to estimate differences in the cardiovascular benefits between both groups. Could an increase in HDL levels be more beneficial at the CV level than a decrease in LDL level? I recommend that they provide the HDL / LDL ratio and Apo A / Apo B which may offer additional data on the possible CV protection of the intervention

Reviewer 3 Report

The research article untitled “Fermented Soy Product Q-Can Natural Reduces Total and LDL Cholesterol: A Randomized Controlled  Crossover Trial” in my opinion do not have enough quality for its publication in a first quartile journal in the field of Nutrition an Dietetic as Nutrients. My decision is based on the lack of novelty of the trial and the not proper analyze of the results from the nutrition and dietetics view.

 MAJOR COMMENTS

The authors propose as hypothesis of the study the scarce amount of researches investigating the effects of a fermented soy product which does not contain active cultures. However, according to the results  and discussion presented in the article, there is no evidence that support the difference interpretation of researches with fermented soy product including active cultures or not. With the purpose to highlight the non-probiotic soy product effects on CVD, a control group with probiotic soy product would have been appropriate.  In the line of improving the experimental design of the research, the are some important factors described in the discussion that must be taken into account. A limitation stated in the manuscript is the larger proportion of females than males. Female participant inclusion would be sufficient to reach 21 participants (n calculated as necessary by the authors line 189). However, there is other important aspect, the age of the participant. Table 2 shows that the mean age of participants was around 50 years old. The problem is that in the case of females physiological, hormones variation occur in this age period. For this reason, physiological stage (premenopause, perimenopause and postmenopause) must be concreted as inclusion criteria of the trial. Otherwise, the effect of soy product in not comparable among the female participants in the intervention group. Indeed, authors give importance to this issue in some parts of the discussion (lines 337-342 of 346-347 for instance).

The most relevant result from the research is the change after the intake of the functional food (fermented soy product) on LDL cholesterol. I do not consider total cholesterol change as appropriate, due to the fact that the variation is conditioned by the LDL and HDL cholesterol reduction, and HDL cholesterol reduction is not a positive effect. Authors do not pay attention to nutritional and dietetics aspects to give respond to the change. According to the supplementary table 2, cholesterol consumption is higher in control group than in soy fermented group. Although the result could explain the LDL variation, other key factors could be found in the diet. The proportion of animal and vegetable protein, the intake of vitamins related to homocysteine B12, B6 and folic acid are a clear example.

Supplementary table 2 shows that energy intake is similar in the two groups, but not information is provided about energy balance. A higher value of energy expenditure in control group could be responsible for the observed LDL cholesterol boost.  In fact, this effect could be assigned to polyphenols content of brown rice, as proposed by the authors (line 354). Taking into account this information and that the nutritional and fiber content between fermented soy and germinated brown rice is different, I feel that the use of germinated brown rice is not a limitation of the sturdy. It is a point that invalidated the control group. With regard to the table 1, sugar and soluble dietary fiber content is necessary to describe. Furthermore, nutrient composition of both products must be calculated in the same form. I mean, the same analytical techniques used by USDA to quantify nutrients and fiber or isoflavones must be used for fermented soy evaluation.

In order to avoid any conflict of interest, authors shouldn’t  use the explicit reference to Q-Can Natural. Q-Can Natural name should be used exclusively in the 2.3. section, to explain the provider  of the soy Fermented product. I suggest as title of the manuscript.  A Non-Probiotic Soy Product Reduces Total and LDL Cholesterol: A Randomized Controlled  Crossover Trial

conclusion section is too expeculative (lines 377-381)

MINOR COMMENTS

Please try to use more recent references (for instance doi: 10.1038/ejcn.2016.77.)

Line 82. Define BMI.

Lines 99-103. What is the database used for nutrient composition of participants’ diet.

Line 124. Can be considered estrogen therapy an exclusion criteria?

Different terms are used in tables and figures for the fermented soy (fermented soy, FSP, Soy Fermented). Try to use always the same.

2.4 section. Laboratory measurement should be described in detail in order to be repeated by others. How were measeured total cholesterol (TC), HDL cholesterol, triglycerides, ApoA1, ApoB, glucose and insulin? Please describe the procedure used in liquid chromatography-mass spectroscopy (line 186)

Lines 219-221. This is a part of the M&M.

Line 228. There is a difference in carbohydrates intake.  

Round 2

Reviewer 3 Report

I really appreciate the author’s effort for including some of my comments in the new version of the manuscript. However, authors must sort an essential issue out. 

The authors response to one of my mayor comments that Dietary cholesterol change was an exception and suggests a possible effect of seasonality.  Due to the crucial relationship between diet cholesterols and blood cholesterol, its variation can not be taken by chance. Dietary cholesterol difference, it is an important result that must be state in results section. Furthermore, the authors have to justify diet cholesterol change during the discussion (they can postulate seasonability effect with references) or admit as limitation.

From my view, it is compulsory to include these two points in the manuscript.
